# Investigating Maternal Stress, Depression, and Breastfeeding: A Pregnancy Risk Assessment Monitoring System (2016–2019) Analysis

**DOI:** 10.3390/healthcare11121691

**Published:** 2023-06-09

**Authors:** Shubhecchha Dhaurali, Vickie Dugat, Tayler Whittler, Shikhar Shrestha, Marwah Kiani, Maria Gabriela Ruiz, Iman Ali, Courtney Enge, Ndidiamaka Amutah-Onukagha

**Affiliations:** 1Department of Community Health, Tufts University, 419 Boston Avenue, Medford, MA 02155, USA; 2Maternal Outcomes for Translational Health Equity Research (MOTHER) Lab, Center for Black Maternal Health and Reproductive Justice, Tufts University School of Medicine, 136 Harrison Avenue, Boston, MA 02111, USAcourtney.enge@tufts.edu (C.E.);; 3Maternal Health Epidemiology and Data Synthesis Unit, Center for Black Maternal Health and Reproductive Justice, Tufts University School of Medicine, 136 Harrison Avenue, Boston, MA 02111, USA; shikhar.shrestha@tufts.edu; 4Department of Public Health and Community Medicine, Tufts University School of Medicine, 136 Harrison Avenue, Boston, MA 02111, USA

**Keywords:** breastfeeding, psychosocial stress, depression, PRAMS, pregnancy, maternal mental health

## Abstract

Breastfeeding is invaluable for postpartum physical healing and mental wellbeing, but psychosocial stress and depression impede such recovery processes. To inform future interventions and policies, associations between breastfeeding, maternal stress, and depression were examined. Data from the Pregnancy Risk Assessment Monitoring System (PRAMS) were analyzed (2016–2019). Logistic regression models were used to calculate adjusted odds ratios with 95% confidence intervals. Of the total sample (*n* = 95,820), approximately 88% of participants attempted breastfeeding. Our findings indicate that participants who experienced any form of stress had a slightly higher likelihood of breastfeeding compared to those without stress. Specifically, partner-related and financial-related stressors were significantly associated with increased odds of breastfeeding. However, no significant associations were observed trauma-related or emotional-related stressors and breastfeeding. Additionally, no significant association was found between depression at different stages (preconception, prenatal, and postpartum) and breastfeeding. A significant interaction effect was noted between having experienced any of the 13 stressors and Black race/ethnicity on breastfeeding odds. Similarly, significant interaction effects were observed between partner-related, trauma, financial, or emotional stressors and Black race/ethnicity. These findings emphasize the importance of considering various factors when promoting breastfeeding in diverse populations, and screening for psychosocial stress during postpartum visits. Our study recommends tailoring breastfeeding interventions to address the needs of Black mothers which could significantly improve maternal health and breastfeeding outcomes.

## 1. Introduction

As one of the leading complications in pregnancy and postpartum, maternal mental health is a growing public health concern. The World Health Organization (WHO) defines mental health as a state of wellbeing in which a mother is aware of her own skills, capable of handling everyday pressures, capable of performing meaningful work, and able to give back to her community [1,2]. Approximately one in five mothers experience a mental health illness during the perinatal period, the time during pregnancy or in the year after birth [3]. As the backbone of many families, mothers overwhelmingly carry the responsibility and stress burden of household management. Birthing people who experience a specific type of stress, psychosocial stress, derived from social situations, are at risk for adverse health outcomes due to implications that stress has on the body [4]. For example, psychosocial stress is associated with prolonged excessive neurobiological stress which may result in the increase in free cortisol in the body [5]. Increased levels of cortisol are associated with adverse health implications. In fact, previous literature has reported that experiencing high levels of stress during pregnancy is associated with an increased risk for preterm birth, infant mortality, postpartum depression, and preeclampsia [4,6], all of which mothers of color are disproportionately predisposed to due to the effects of systemic racism [6].

Stressful life events related to motherhood include birth-related complications and motherhood stress, which are often precursors to self-doubt and insecurity and can eventually induce postpartum depression [7]. Overall, these stressors can cause constant worrying and lead to adverse physiological stress responses and lack of energy in birthing persons, which can result in slow digestion, fatigue, and high blood pressure [8]. The spectrum of symptoms may manifest in postpartum depressive episodes that mothers experience, which can lead to detrimental impacts on maternal and infant health alike. Such examples include reduced self-esteem, adverse maternal–infant interactions, not seeking out preventative healthcare treatment, and breastfeeding impediment [9].

Depressive symptomatology appearing in the first year following labor and delivery is known as postpartum depression [10]. It is a major depressive maternal mental and emotional health disorder recognized in the Fifth Edition Text Revision of the American Psychiatric Association’s Diagnostic and Statistical Manual of Mental Disorders (DSM-5-TR) [11,12,13]. The imbalance of hormones after birth may trigger a spectrum of postpartum depressive symptoms including insomnia, fatigue, sadness, hardship with maternal-infant bonding, self- and infant harm ideation, among other impacts [14]. The added experiences of being sleep-deprived and overwhelmed by adjusting to postpartum life may also induce stress that can exacerbate postpartum depressive symptoms [15]. While there have been numerous studies on factors that impact breastfeeding initiation and duration, few studies have assessed the influence of maternal psychosocial stress and maternal mental health on breastfeeding. Many studies point to an association between maternal mental health and breastfeeding [16,17,18]; however, the causal pathway is unclear. It is uncertain if a decrease in breastfeeding causes an increase in mental health concerns such as postpartum depression or if an increase in such mental health concerns causes a decrease in breastfeeding.

This study investigated the association between categorical maternal stressors experienced during pregnancy, maternal mental health disorders, and breastfeeding among mothers in the United States. It is important to examine these relationships and add to the existing literature regarding breastfeeding to inform future health screenings and/or interventions for maternal psychosocial stress and/or postpartum depression to promote physical, mental, and social wellbeing.

## 2. Materials and Methods

### 2.1. Data Source

This present study conducted a secondary data analysis using the Pregnancy Risk Assessment and Monitoring Systems (PRAMS) data from 2016–2020 (Phase 8). PRAMS is a population-based standardized surveillance system administered by the Centers for Disease Control and Prevention (CDC), as well as state, local, and territorial health departments [19]. PRAMS data represent approximately 81% of all births in the United States, and collects self-reported data related to women’s experiences and attitudes during preconception, pregnancy, and postpartum experiences and behaviors among women who have had a recent live birth [19]. A detailed description of PRAMS study design and sampling methods are available elsewhere [19]. Please note that in this study, the usage of the terms “women” and “mothers” is intended to be inclusive of all birthing individuals, although PRAMS exclusively uses the term “women” and does not specify further.

### 2.2. Analytic Sample

The study sample consisted of women who participated in PRAMS Phase 8 (2016–2020, *n* = 202,745), although, due to the substantial influence of the COVID-19 pandemic on maternal demographics, stress, mental health, and potentially breastfeeding practices, this study excluded the year 2020 (*n* = 39,717 (19.6%)) from the final sample. PRAMS sites for which data on psychosocial stress from question P19 were not available were also excluded (*n* = 60,798 (30%)). Those sites with core and standard maternal stress and mental health questions included the following 30 sites: Alaska (AK), Alabama (AL), Colorado (CO), Connecticut (CT), Delaware (DE), Florida (FL), Georgia (GA), Iowa (IA), Illinois (IL), Indiana (IN), Kansas (KS), Kentucky (KY), Louisiana (LA), Massachusetts (MA), Maine (ME), Michigan (MI), Minnesota (MN), Missouri (MO), Mississippi (MS), North Carolina (NC), Nebraska (NE), Oklahoma (OK), Oregon (OR), Pennsylvania (PA), Texas (TX), Utah (UT), Washington (WA), Wisconsin (WI), Wyoming (WY), and New York City (NYC). The sites included encompass states and a major city located in various regions of the country.

The study was approved by the federal CDC PRAMS and the PRAMS grantees at the 30 previously listed sites. This study was deemed Not Human Subjects research and determined to be exempt from IRB approval because it was a secondary analysis of a publicly available de-identified dataset.

### 2.3. Exposures and Outcome

#### 2.3.1. Breastfeeding Outcome

The primary outcome of interest, breastfeeding, was measured using participants’ responses to the following questions from the PRAMS data set: “Did you ever breastfeed or pump breast milk to feed your new baby, even for a short period of time?” and “Are you currently breastfeeding or feeding pumped milk to your new baby?” [20]. Participants who answered “yes” to both questions were coded “yes” for breastfeeding, and those who answered “no” to both questions were coded as “no”. Participants who answered “yes” to only one of two questions were coded as breastfeeding as well.

#### 2.3.2. Psychosocial Stress Exposures

One of two main exposure variables was psychosocial stress. Stressors were measured based on mothers’ “yes” or “no” responses to the following 14 statements: (1) a close family member was very sick and had to go into the hospital, (2) separated or divorced from my husband or partner, (3) * moved to a new address, (4) was homeless, (5) husband or partner lost his job, (6) lost my job even though I wanted to go on working, (7) argued with my husband or partner more than usual, (8) husband or partner said he didn’t want me to be pregnant, (9) had a lot of bills I couldn’t pay, (10) husband, partner, of I experienced cut pay or hours, (11) husband or partner or I went to jail, (12) someone very close to me had a problem with drinking or drugs, (13) partner was away due to military deployment or extended work-related travel, and (14) someone very close to me died [20]. Based on previous studies, stressful life events were categorized into four groups excluding stressor 3: partner-related stress (2, 7, 8, 13), trauma-related stress (4, 10, 11, 12), financial stress (5, 6, 9), and emotional stress (1, 14) [4,21,22,23]. * Question 3 was excluded because this question could be interpreted either as a positive or negative experience and/or consequence [24]. Participants who responded “yes” to any of the 13 stressors were coded as “yes” for psychosocial stress and their respective stressor categories.

#### 2.3.3. Depression Exposures

Exposure variables were maternal mental health conditions throughout the birthing process. More specifically, the variables were preconception depression, prenatal depression, postpartum depression. Preconception depression was measured based on PRAMS core question 4c, “During the 3 months before you got pregnant with your new baby, did you have [depression]?” [20] “Yes” responses were coded as “yes” for preconception depression, and “no” responses as “no”. Prenatal depression, or depression experienced during pregnancy, was measured based on PRAMS core question 18c, “During your most recent pregnancy, did you have [depression]?” [20] Similarly to preconception depression, “yes” and “no” responses were coded as such. PRAMS questions 48 and 49 ask participants whether they have experienced postpartum depressive symptoms, and such symptoms act as surrogates for postpartum depression as the condition is not formally diagnosed [4]. These questions are: “Since your new baby was born, how often have you felt down, depressed, or hopeless?” and “Since your new baby was born, how often have you had little interest or little pleasure in doing things you usually enjoyed?” [20]. Responses were based on the 5-item Likert scale: never (1), rarely (2), sometimes (3), often (4), always (5). Similar to what was performed in a previous study, participants who responded “always (5)” or “often (4)” were coded as “yes” and those who responded “never (1)”, “rarely (2)”, or “sometimes (3)” were coded as “no” [4]. Participants who responded “yes” to both questions were coded as “yes” for postpartum depression and those who responded “no” to both questions were coded as “no”.

### 2.4. Covariates

Covariates were included to account for sociodemographic characteristics of mothers based on significance. Significance was determined based on a Chi-square test, where the significance was *p* ≤ 0.01 (Bonferroni correction). The sociodemographic factors that were significant and selected were maternal age, education, marital status, race/ethnicity, household income, prenatal health insurance, parity, and pregnancy intention.

Marital age was grouped into three categories based on participants’ responses on their age. For participants younger than or 17 years of age, between 18 and 19 years old, and 20–24 years, they were grouped as younger than or of 24 years of age. For individuals who responded being 25–29 or 30–34, they were grouped as 25–34 years old. Lastly, for those 35–39 years of age or older, they were grouped into the 35 or older age group. Maternal education was dependent on participants’ responses to the number of years they were in school. Those who reported between 0 and 11 years of schooling were coded as having less than a high school education, those who reported exactly 12 years as completing high school or its equivalent, those who reported 13–15 years as having completed some college, and those who reported more than or equal to 16 years having completed at least a college degree. Marital status was coded either as married or other as PRAMS reported it. Maternal race was categorized into six distinctions: (1) “Black, non-Hispanic” (2) “White, non-Hispanic” (3) “Hispanic” (4) “American Indian/Alaskan Native, non-Hispanic” (5) “Asian, non-Hispanic”, and (6) “Other, non-Hispanic”. Participants who answered “yes” to if their race was “Black” and “no” to if they were “Hispanic” were coded as non-Hispanic Black. This was similar for non-Hispanic White, Asian, American Indian/Alaskan Native, and non-Hispanic Other identifying participants. Participants who responded “yes” to Hispanic were represented as Hispanic.

Income was grouped based on the participants’ answers to PRAMS core question 50: “During the 12 months before your new baby was born, what was your yearly total household income before taxes? Include your income, your husband’s or partner’s income, and any other income you may have received” [20]. Responses one to three were coded as earning between USD 0 and USD 24,000, responses four–seven as USD 24,001–USD 48,000, responses eight–ten as USD 48,001–USD 73,000, and responses eleven and twelve as more than USD 73,000. Insurance status was coded based on participants’ answers to core question 10: “During your most recent pregnancy, what kind of health insurance did you have for your prenatal care?” [20]. Participants who responded “yes” for either insurance paid by job, parent, or Healthcare Exchange in their state were coded as having private health insurance. Participants who responded “yes” to being insured by their state Medicare were coded as having Medicaid public health insurance, and those who reported “none” were coded as having no health insurance.

Parity was categorized into none, one, or two or more based on the number of previous live births participants reported. Participants who reported zero were coded as one, those who reported one as one, and participants who reported 2, 3–5, or more than 6 into the two or more category. Pregnancy intention was measured through PRAMS Phase 8 core question 12: “Thinking back to just before you got pregnant with your new baby, how did you feel about becoming pregnant?” [20]. Response choices included the following: (1) I wanted to be pregnant later, (2) I wanted to be pregnant sooner, (3) I wanted to be pregnant then, (4) I didn’t want to be pregnant then or at any time in the future, (5) I wasn’t sure what I wanted [20]. In accordance with the CDC definition of unintended pregnancy intention, participant responses were coded into the three different following groups: (a) intended (1, 2, 3), (b) unintended (4), and (c) unsure (5) [4,25].

### 2.5. Data Analysis

Frequencies and percentages were used to describe sample characteristics. Chi-square tests were used to determine significant differences across row or column groups at *p* < 0.01. Logistic regression procedures were applied to determine the association between psychosocial stressors, postpartum depression, and breastfeeding adjusting for all covariates. Adjusted odds ratios (aORs), 95% confidence intervals (CI), and *p*-values were calculated for each of the independent variables. Significant variables at the value of *p* < 0.01 level in the bivariate analyses were included in the multivariable logistic regression models. To address this issue of missingness for variables with more than 5% of missing data, we performed sensitivity analyses to compare the estimates of our outcome variables when we imputed the missing observations. In the logistic regressions, our total was *n* = 75,419 due to automatically omitted missing variables via the programming. We used the ologit function to impute income missingness and the mlogit function to impute insurance missingness with the raw site and covariate variables as predictors. We used the imputed variables in our regression models as predictors, and the findings are shown in Appendix A. Data analysis was conducted accounting final survey weights using Stata Corp, LLC Version 17.0.

## 3. Results

### 3.1. Sample Characteristics

Maternal characteristics by stressor types (any of the 13 categorized stressors, partner-related, trauma-related, finance-related, and emotion-related) are shown in Table 1. Among the participants, 23.5% were ≤24 years old, 58.9% were between 25 and 34 years old, and 17.7% were ≥35 years old. Approximately 11.4% had less than a high school diploma, 24.4% had a high school diploma, 27.8% had some college education, and 36.4% had completed college. The majority of participants were married (62.2%), non-Hispanic White (henceforth referred to as White) (61.6%) earned more than a household income of USD 73,000 (33%) and possessed private insurance (53.7%). The majority of the sample population intended their pregnancy (44.2%) and identified as first-time mothers (38.6%).

The highest percentage of participants experiencing any of the stressor categories was the age group of 25–34 years (any stressor—57% vs. partner—53.4% vs. traumatic—52.8% vs. financial—55.2% vs. emotional—58.2%). Participants who reported a household income of USD 0–USD 24,000 also were among the majority to experience all stress categories (38% vs. 44.8% vs. 52.2% vs. 48.9% vs. 34.6%). Married participants had lower frequencies of experiencing partner-related (46%), trauma-related (38.9%), and finance-related (47.1%) stressors compared to those in the “other” category. Participants with Medicaid public health insurance had higher frequencies of experiencing any (50.9%), partner-related (58.5%), traumatic (65.1%), and financial (63.5%) stressors compared to those with private insurance. Participants with unintended pregnancies had higher frequencies of experiencing any stressor (44.3%), partner-related stressor (48.8%), traumatic stressor (47%), financial (46.5%), and emotional stressor (43.3%) compared to those with intended or unsure pregnancies.

Maternal characteristics by reported experiences of depression (during preconception, prenatal, and postpartum periods) are shown in Table 2. Participants aged ≤24 had the highest frequencies in all three periods (19.6% vs. 19.1% vs. 20.5%), followed by participants aged 25–34 years. Participants with completed college education had the lowest frequencies in all three periods (8.1% vs. 7.4% vs. 9.5%), while those with a high school diploma had the highest. American Indian/Alaskan Native participants had the highest frequency reporting preconception and prenatal depression (24% vs. 21.2%), and Black participants reported the highest frequencies of postpartum depression (19.9%). Participants in the higher income categories had lower frequencies in all three depression periods compared to those in the lower income categories. Participants with Medicaid insurance had higher frequencies in all three depression periods (19.6% vs. 19.3% vs. 18.8%) compared to those with private or no insurance. Those who reported unsure pregnancy intention or had two or more previous births had higher frequencies in all three depression periods.

Maternal characteristics by breastfeeding are shown in Table 3. Participants aged ≤24 years had the lowest percentage of breastfeeding (83%), followed by those aged 25–34 (88.9%) and ≥35 (90%) years. Participants with less education (high school diploma or below) had lower percentages of breastfeeding. Asian participants had the highest percentage of breastfeeding (93.3%), followed by Other (92.6%), Hispanic (91.5%), American Indian/Alaskan Native (90%), and White (88.6%). Black participants had a relatively lower percentage of breastfeeding (77.8%) compared to the other races/ethnicities. Participants with higher household incomes (>USD 73,000) had a higher percentage of breastfeeding (94.9%), while those in lower income categories had lower percentages. Participants with private insurance had a higher percentage of breastfeeding (93.1%) compared to those with Medicaid (81%) or no insurance (90.8%). Participants with intended pregnancies had a higher percentage of breastfeeding (90.4%) compared to those with unintended (87%) or unsure (81.7%) pregnancies. Participants with no children had the highest percentage of breastfeeding (90.7%), followed by those with one child (87.5%) and two or more children (83.9%).

### 3.2. Association between Psychosocial Stressors and Breastfeeding

Figure 1 presents the adjusted odds ratios (aOR) and 95% confidence intervals (95% CI) representing the association between different psychosocial stressor categories and the outcome variable of breastfeeding. The findings indicate that participants who experienced any form of stress had a slightly higher likelihood of breastfeeding compared to those without any stress (aOR = 1.12, 95% CI: 1.04–1.21, *p* ≤ 0.01). Similarly, participants facing stress related to their partner exhibited slightly increased odds of breastfeeding compared to those without partner-related stress (aOR = 1.11, 95% CI: 1.02–1.20, *p* ≤ 0.01). While participants who experienced trauma had slightly elevated odds of breastfeeding in comparison to those without trauma-related stress (aOR = 1.11, 95% CI: 1.00–1.22), this association did not reach statistical significance. Conversely, participants experiencing financial stress demonstrated significantly higher odds of breastfeeding when compared to those without financial stress (aOR = 1.17, 95% CI: 1.08–1.26, *p* ≤ 0.001). However, the presence of emotional stress did not exhibit a significant association with the outcome variable when compared to individuals without emotional stress (aOR = 1.04, 95% CI: 0.97–1.12). Please reference Appendix A for the sensitivity analysis with imputed variables.

### 3.3. Association between Breastfeeding and Depression

The logistic regression analysis in Figure 2 explored the association between breastfeeding and depression at different stages: preconception, prenatal, and postpartum. The findings suggest that there is no significant association between breastfeeding and depression. The odds ratios for each stage (preconception, prenatal, and postpartum) were close to one, indicating that the odds of breastfeeding were not significantly different between individuals with and without depression. The 95% confidence intervals for all three stages included one, further supporting the lack of significant association. These results suggest that depression may not have a substantial impact on breastfeeding behavior during the preconception, prenatal, and postpartum periods. Please reference Appendix A for the sensitivity analysis with imputed variables.

### 3.4. Association between Breastfeeding and Race

The association between breastfeeding and race among different racial/ethnic groups is presented in Figure 3. The findings indicate significant associations between race and breastfeeding. Compared to the reference group (White individuals), Black individuals had lower odds of breastfeeding (aOR = 0.82, 95% CI: 0.75–0.90, *p* ≤ 0.01). In contrast, Hispanic individuals had substantially higher odds of breastfeeding (aOR = 2.66, 95% CI: 2.35–3.00, *p* ≤ 0.001). Similar trends were observed for American Indian/Alaskan Native individuals, Asian individuals, and individuals categorized as “Other”. These groups exhibited increased odds of breastfeeding, with aORs of 1.93, 1.60, and 2.24, respectively (all significant at *p* ≤ 0.001). Please reference Appendix A for the sensitivity analysis with imputed variables.

### 3.5. Interaction between Maternal Psychosocial Stressors and Race on Breastfeeding

The interaction between maternal psychosocial stressors and race/ethnicity on breastfeeding is shown in Table 4. We observed a significant interaction effect between any stress and Black race/ethnicity on the odds of breastfeeding after controlling for other confounders (aOR: 1.27, 95% CI: 1.07–1.51, *p* ≤ 0.01). Similarly, we observed a significant interaction effect between partner-related (aOR: 1.32, 95% CI: 1.14–1.54, *p* ≤ 0.001), trauma (aOR: 1.34, 95% CI: 1.16–1.55, *p* ≤ 0.001), financial (aOR: 1.26, 95% CI: 1.08–1.47, *p* ≤ 0.01), or emotional (aOR: 1.38, 95% CI: 1.18–1.60, *p* ≤ 0.001) stressors and Black race/ethnicity. All other interaction effects between race–ethnicity categories and stress categories were not statistically significant. Please reference Appendix A for the sensitivity analysis with imputed variables.

## 4. Discussion

The results of this population-based study of mothers who gave birth between 2016–2019 demonstrate a strong association between maternal stress and breastfeeding. Although previous studies have indicated a negative association between breastfeeding and psychosocial stressors [26,27,28], the findings of this study allude to a somewhat positive association. This study found that mothers who experienced any, partner, and financial stressors during pregnancy were more likely to breastfeed when controlled for maternal age, level of education, marital status, race, parity, income, insurance, and pregnancy intention. Our findings are not consistent with the literature which reports that maternal stress impedes breastfeeding processes. Previous studies have shown mothers who experience psychosocial stress and decreased levels of social support also reported early discontinuation of breastfeeding practices [29,30]. Our study difference partially could have been due to factors such as the questions used to measure and categorize stressors, and the covariates used in the logistic regression model.

The findings of this study indicate a lack of association between depression and breastfeeding. Notably, our sensitivity analysis revealed no substantial changes in the estimated association between the predictor variables and breastfeeding across all logistic regression models. These results further support the initial finding that there is no significant relationship between depression and breastfeeding. Our findings were similar to those of some studies found in the literature that expressed depression experienced during pregnancy does not predict breastfeeding initiation and intention [16]. The systematic review’s findings were unclear regarding whether breastfeeding would be considered a mediating factor between pregnancy and postpartum depression [16]. Additionally, Kroll and Grossman (2018) asserted that prenatal depression is associated with decreased breastfeeding practices, implying that while breastfeeding might impact maternal mental health, mental health may also impact breastfeeding likelihood [17]. Another systematic review interestingly found that societal pressures to breastfeed also impacted maternal mental health [18]. The study stated that when women viewed breastfeeding as integral to motherhood, they had a higher chance of continuing to breastfeed beyond 3 months compared to mothers who did not view breastfeeding in such a way. Among the many biological and physical factors that put mothers at risk of experiencing postpartum depression, behavioral and environmental risk factors that predispose mothers to postpartum depression include a negative appraisal of pregnancy, lack of social support, stressful life events, abuse, preconception depression, anxiety, unsatisfying marriage, and prior experience with adverse mental health outcomes [31]. While these factors play a role in breastfeeding behaviors, some risk factors are not evident until after childbirth and require further postpartum health evaluations. However, several influencing factors can be recognized based on past medical history and throughout preconception care checkups. It is critical that mothers have access to quality equitable postpartum care. Through holistically monitoring postpartum recovery, providers can better identify postpartum depressive symptoms and intervene by providing the appropriate care while providing mothers who want to breastfeed with the support they need.

Consistent with previously published research, the findings of this study add to the literature that has indicated an association between maternal race, stressors, and breastfeeding [32,33,34]. We found a significant interaction effect between any stress and Black race/ethnicity on breastfeeding odds. Similarly, significant interaction effects were observed between partner-related, trauma, financial, or emotional stressors and Black race/ethnicity. Financial stress may be associated with an increased odds of breastfeeding due to various factors. Breastfeeding is a cost-effective method of feeding an infant compared to the expenses associated with buying formula milk, bottles, and other feeding equipment. Financially stressed individuals or families may choose breastfeeding as a means to save money on infant feeding costs. Moreover, breastfeeding offers health advantages to both the mother and the baby, potentially lowering medical expenses related to infant illnesses and postpartum complications. Consequently, individuals experiencing financial stress may be more inclined to breastfeed in order to provide affordable and beneficial nutrition for their babies.

Beyond socioeconomic barriers, medical discrimination as it relates to historical and modern-day interactions that Black women have faced within the healthcare system has led to poor healthcare access, utilization, and treatment [35]. Acknowledging the historical trials that Black women have faced and continue to face is salient in understanding the factors that have shaped perceptions and attitudes toward breastfeeding within this community. Many Black women have associated trauma and discomfort with healthcare institutions as a result of being silenced, oppressed, mistreated, and violated among many other experiences, and those exposures have discouraged them from continually seeking and utilizing healthcare services [36]. Despite these challenges, Black women often demonstrate a strong cultural preference for breastfeeding and a high initiation rate; however, exposure to adverse healthcare services could subsequently influence breastfeeding practices and decrease breastfeeding duration rates among Black mothers. The association between being Black, experiencing stress, and breastfeeding may be attributed to multiple factors. Cultural beliefs and norms within Black communities may place a strong emphasis on the importance of breastfeeding. This cultural influence can create a supportive environment that encourages Black women to initiate breastfeeding despite the presence of stressors. This highlights the resilience and cultural significance of breastfeeding within Black communities. However, more research is required to comprehensively examine the interplay between stress, race, and breastfeeding outcomes. It is critical to consider these concerns when creating interventions because without efforts that cater to the unique needs of Black women, there will be no feasible pathway for effective solutions.

Although no one incident is solely responsible for maternal mental health conditions and the reasons women experience varying events of stress during pregnancy, it is important to explore the unique barriers that mothers encounter while on the journey of motherhood and how that impacts their ability to breastfeed if they so desire. Not only do mothers endure biological changes during pregnancy, but they also must continue to navigate the responsibilities of life. The implications of these findings for policy and practice are multi-faceted. Policymakers and healthcare providers should consider the socio-economic stress experienced during pregnancy and the cultural factors that influence breastfeeding practices. Developing targeted interventions and support systems for individuals facing financial stress such as providing access to breastfeeding education, lactation support, and resources at low to no costs can help mitigate the barriers and increase breastfeeding rates. Addressing racial disparities in breastfeeding outcomes requires a comprehensive approach. Policies should focus on improving access to quality healthcare, promoting culturally sensitive breastfeeding education, and providing support services specifically tailored to the needs of Black communities. Recognizing and addressing the unique challenges faced by Black women, such as racial discrimination and social inequalities, is crucial for promoting breastfeeding equity.

The findings of this study should be considered with several limitations and strengths. The essential limitations of the study were: (1) the self-reported nature of the data, which increases the potential for social desirability bias, selection bias, and recall bias; (2) grouping respondents into the five racial/ethnic groups, which is not representative of the many different cultural and racial identities that exist in the United States; (3) sites that did not include our exposure or outcome questions from the PRAMS questionnaire were excluded, and (4) missing data for income and insurance categories which could affect the estimates of the odds of breastfeeding. However, the sensitivity analyses showed that there is no significant change in the aOR estimates of the association between our predictor variables (stressors and depression) and outcome variable (breastfeeding) for all logistic regression models. Despite the limitations, the study had several strengths that included: (1) evidence based upon the PRAMS data which includes a large sample of women in the United States; (2) standardized and consistent data collection process; and (3) a comprehensive list of variables in the dataset, which allowed for adjustment for potential confounders.

## 5. Conclusions

Maternal mental health is an escalating global crisis, and this study underscores the significance of comprehending the association between psychosocial stressors, maternal depression, and breastfeeding while also identifying potential racial disparities. After controlling for various demographic variables, the study findings reveal that women who experienced any of the 13 partner-related and/or financial psychosocial stressors were slightly more likely to breastfeed than those who did not, which opposes the findings of the general literature. Further investigation using a randomized controlled study design is encouraged to determine the exact association. The distress that mothers experience emphasizes the critical role of social support in improving maternal well-being and facilitating optimal infant development [37]. Although our study did not find a significant association between depression and breastfeeding practices, maternal mental health throughout pregnancy and the postpartum periods is critical in ensuring maternal and infant health. Thus, healthcare providers must remain vigilant in screening and monitoring postpartum depressive symptoms and psychosocial stressors. Screening programs allow for the identification of symptoms and enable the provision of appropriate interventions and support for breastfeeding mothers, particularly Black mothers who have experienced discomfort and trauma within the healthcare system. Furthermore, tailored breastfeeding programs and interventions are necessary to promote equitable outcomes. Future studies must address the impact of psychosocial stress on the duration of breastfeeding and the effects of the COVID-19 pandemic on maternal mental health.

## Figures and Tables

**Figure 1 healthcare-11-01691-f001:**
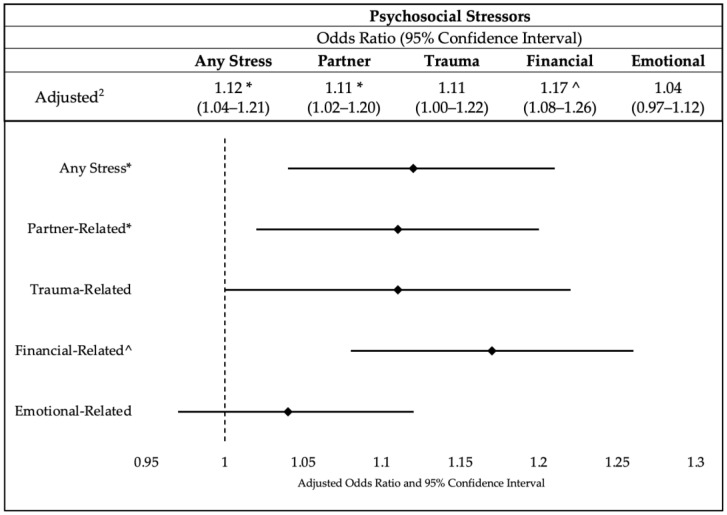
Association between psychosocial stressors and breastfeeding (*n* = 75,419) Note: * Significance at *p* ≤ 0.01; ^ Significance at *p* ≤ 0.001.

**Figure 2 healthcare-11-01691-f002:**
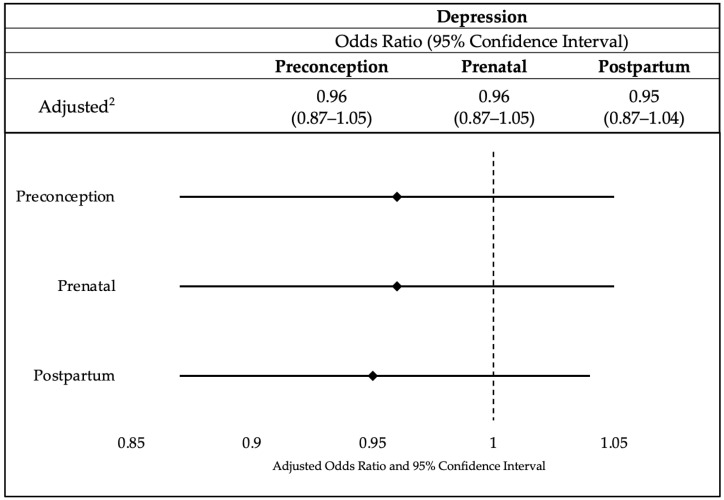
Association between breastfeeding and depression (*n* = 75,419).

**Figure 3 healthcare-11-01691-f003:**
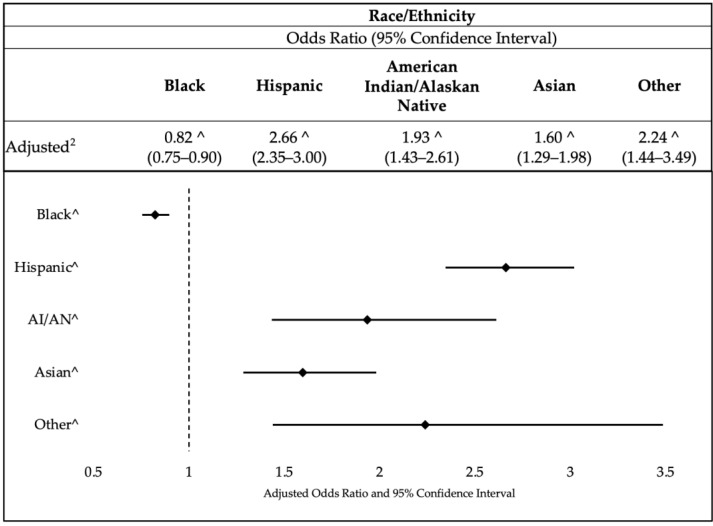
Association between breastfeeding and race (*n* = 75,419) Note: ^ Significance at *p* ≤ 0.001.

**Table 1 healthcare-11-01691-t001:** Maternal Characteristics by Stressors (*n* = 95,820).

Characteristics	Total	Any Stressor	Partner	Traumatic	Financial	Emotional	*p*-Value ^‡^
Age (in years)		Yes (%)					
≤24	22,021 (23.5)	15,020 (27.2)	8069 (32.3)	4626 (35.5)	8439 (30.4)	7440 (25.9)	
25–34	56,490 (58.9)	32,326 (57)	14,221 (53.4)	7360 (52.8)	16,131 (55.2)	16,950 (58.2)	*p* ≤ 0.001
≥35	17,309 (17.7)	9422 (15.8)	4069 (14.4)	1685 (11.8)	4346 (14.4)	4914 (15.9)	
Education							
<High school diploma	11,266 (11.4)	6783 (11.5)	3306 (12.7)	2091 (15.4)	3837 (12.9)	3353 (10.6)	
High school diploma	23,235 (24.4)	15,006 (26.8)	7399 (29)	4424 (32.2)	8682 (30.9)	7327 (25.4)	*p* ≤ 0.001
Some College	28,427 (27.8)	18,714 (31)	9159 (33)	4888 (34.2)	10,481 (34.4)	9530 (30.1)	
Completed college	32,892 (36.4)	16,265 (30.6)	6495 (25.4)	2268 (18.2)	5916 (21.8)	9094 (33.8)	
Marital Status							
Other	37,954 (37.8)	26,533 (44.9)	14,439 (54)	8630 (61.1)	15,693 (52.9)	12,896 (41.2)	*p* ≤ 0.001
Married	57,866 (62.2)	30,235 (55.1)	11,920 (46)	5041 (38.9)	13,223 (47.1)	16,408 (58.8)	
Race/Ethnicity							
Black	19,946 (15.2)	13,497 (17.3)	7378 (21.2)	3269 (17.8)	7932 (20.4)	6815 (16.5)	
White	48,812 (61.6)	28,130 (60.8)	11,837 (56)	6894 (65)	12,938 (55.9)	15,506 (65.2)	
Hispanic	16,462 (17.5)	9506 (17.3)	4404 (18.1)	1868 (13.6)	5283 (19.3)	4123 (14.2)	*p* ≤ 0.001
American Indian/Alaskan Native	4169 (1)	2904 (1.2)	1460 (1.3)	1325 (2.1)	1519 (1.3)	1620 (1.3)	
Asian	5502 (3.8)	2245 (2.5)	1027 (2.6)	241 (0.9)	986 (2.2)	1028 (2.2)	
Other	929 (0.9)	486 (0.8)	253 (0.8)	74 (0.5)	258 (0.9)	212 (0.7)	
Household Income ^†^							
USD 0–USD 24,000	28,227 (32.3)	19,778 (38)	10,482 (44.8)	6416 (52.2)	12,642 (48.9)	9614 (34.6)	
USD 24,001–USD 48,000	17,634 (20.5)	11,151 (22.1)	5184 (22.3)	2480 (20.8)	6386 (25.1)	5408 (20.8)	*p* ≤ 0.001
USD 48,001–USD 73,000	12,036 (14.3)	6884 (14)	2881 (12.7)	1248 (10.4)	3123 (12.4)	3707 (14.6)	
>USD 73,000	24,309 (33)	11,230 (25.9)	4134 (20.2)	1614 (16.6)	3021 (13.6)	6775 (30)	
Prenatal Health Insurance ^†^							
Private	43,653 (53.7)	22,016 (46.5)	8473 (38.9)	3693 (33.4)	8120 (33.6)	12,622 (51.5)	
Medicaid	41,315 (43.4)	28,339 (50.9)	14,559 (58.5)	8661 (65.1)	17,648 (63.5)	13,714 (46.4)	*p* ≤ 0.001
None	2148 (2.9)	1176 (2.6)	483 (2.6)	204 (1.5)	686 (2.8)	517 (2.2)	
Pregnancy Intention							
Intended	41,050 (44.2)	21,245 (38.6)	7988 (31)	4089 (30.7)	9504 (33.6)	11,421 (40.8)	
Untended	39,186 (40.7)	24,985 (44.3)	12,624 (48.8)	6314 (47)	13,291 (46.5)	12,715 (43.3)	*p* ≤ 0.001
Unsure	15,584 (15.1)	10,538 (17.1)	5747 (20.2)	3268 (22.3)	6121 (19.9)	5168 (16)	
Parity							
None	36,817 (38.6) *	21,825 (38.8)	9966 (38.3)	5293 (39.6)	10,532 (36.5)	11,432 (39.4) ^#^	
One	30,356 (33) *	17,581 (32.4)	8108 (31.9)	3964 (30.3)	8725 (31.7)	9075 (32.7) ^#^	*p* ≤ 0.001
Two or more	28,647 (28.5) *	17,362 (28.8)	8285 (29.8)	4414 (30.1)	9659 (31.8)	8797 (27.9) ^#^	

Note: ^†^ Different *n* values for income (*n* = 82,206) and insurance status (*n* = 87,116); ^‡^ denotes significance at *p* ≤ 0.001 across columns; * denotes significance at *p* ≤ 0.01 across column; ^#^ denotes non-significance.

**Table 2 healthcare-11-01691-t002:** Maternal Characteristics by depression (*n* = 95,820).

Characteristics	Preconception	Prenatal	Postpartum	*p*-Value ^‡^
Age (in years)	Yes (%)			
≤24	4427 (19.6)	4443 (19.1)	4494 (20.5)	*p* ≤ 0.001
25–34	7532 (12.6)	7453 (12.1)	7770 (13.1)	*p* ≤ 0.001
≥35	2039 (10.7)	1954 (9.8)	2096 (11.3)	*p* ≤ 0.001
Education				
<High school diploma	2069 (18)	2075 (17.2)	2113 (18.6)	*p* ≤ 0.001
High school diploma	4266 (18)	4317 (17.8)	4343 (18.5)	*p* ≤ 0.001
Some College	4839 (16.3)	4761 (15.6)	4602 (15.9)	*p* ≤ 0.001
Completed college	2824 (8.1)	2697 (7.4)	3302 (9.5)	*p* ≤ 0.001
Marital Status				
Married	7505 (19.4)	7783 (19.6)	7313 (19.4)	*p* ≤ 0.001
Other	6493 (10.6)	6067 (9.6)	7047 (11.5)	*p* ≤ 0.001
Race/Ethnicity				
Black	2628 (11.8)	3173 (14.8)	3909 (19.9)	*p* ≤ 0.001
White	8265 (16.1)	7440 (14.4)	6283 (12.8)	*p* ≤ 0.001
Hispanic	1814 (9.5)	1942 (10)	2247 (14.3)	*p* ≤ 0.001
American Indian/Alaskan Native	916 (24)	862 (21.2)	693 (17.9)	*p* ≤ 0.001
Asian	264 (5.1)	326 (5)	1065 (19.6)	*p* ≤ 0.001
Other	111 (9.9)	107 (9.5)	163 (17.1)	*p* ≤ 0.001
Household Income ^†^				
USD 0–USD 24,000	5977 (20.6)	6076 (20.5)	5868 (20.5)	*p* ≤ 0.001
USD 24,001–USD 48,000	2639 (15)	2596 (14.2)	2762 (15.8)	*p* ≤ 0.001
USD 48,001–USD 73,000	1562 (12)	1466 (11.3)	1460 (11.5)	*p* ≤ 0.001
>USD 73,000	2179 (8.3)	1949 (7.2)	2200 (8.9)	*p* ≤ 0.001
Prenatal Health Insurance ^†^				
Private	4599 (10)	4225 (8.9)	4730 (10.5)	*p* ≤ 0.001
Medicaid	8069 (19.6)	8216 (19.3)	7868 (18.8)	*p* ≤ 0.001
None	211 (8.2)	235 (9)	281 (11.5)	*p* ≤ 0.001
Pregnancy Intention				
Intended	4357 (10.2)	4004 (9)	4622 (11.1)	*p* ≤ 0.001
Untended	6594 (16)	6695 (16)	6797 (16.8)	*p* ≤ 0.001
Unsure	3047 (18.9)	3151 (19)	2941 (18.3)	*p* ≤ 0.001
Parity				
None	5473 (14.2)	5107 (12.7)	5485 (14.7) *	*p* ≤ 0.001
One	4064 (12.8)	4102 (12.7)	4398 (13.8) *	*p* ≤ 0.001
Two or more	4461 (14.8)	4641 (15)	4477 (15) *	*p* ≤ 0.001

Note: ^†^ Different *n* values for income (*n* = 82,206) and insurance status (*n* = 87,116); ^‡^ denotes significance at *p* ≤ 0.001 across rows; * denotes significance at *p* ≤ 0.01 across row.

**Table 3 healthcare-11-01691-t003:** Maternal Characteristics by breastfeeding (*n* = 95,820).

Characteristics	Breastfeeding		
	No (%)	Yes (%)	*p*-Value ^‡^
Age (in years)			
≤24	3570 (17)	18,451 (83)	*p* ≤ 0.001
25–34	6137 (11.1)	50,353 (88.9)	*p* ≤ 0.001
≥35	1797 (10)	15,512 (90)	*p* ≤ 0.001
Education			
<High school diploma	2441 (21.4)	8825 (78.6)	*p* ≤ 0.001
High school diploma	4449 (19.7)	18,786 (80.3)	*p* ≤ 0.001
Some College	3214 (12.2)	25,213 (87.8)	*p* ≤ 0.001
Completed college	1400 (4.6)	31,492 (95.4)	*p* ≤ 0.001
Marital Status			
Married	7366 (19.8)	30,588 (80.2)	*p* ≤ 0.001
Other	4138 (7.7)	53,728 (92.3)	*p* ≤ 0.001
Race/Ethnicity			
Black	4274 (22.2)	15,672 (77.8)	*p* ≤ 0.001
White	5028 (11.4)	43,784 (88.6)	*p* ≤ 0.001
Hispanic	1388 (8.5)	15,074 (91.5)	*p* ≤ 0.001
American Indian/Alaskan Native	406 (10)	3763 (90)	*p* ≤ 0.001
Asian	344 (6.7)	5158 (93.3)	*p* ≤ 0.001
Other	64 (7.4)	865 (92.6)	*p* ≤ 0.001
Household Income ^†^			
USD 0–USD 24,000	5850 (20.9)	22,377 (79.1)	*p* ≤ 0.001
USD 24,001–USD 48,000	1916 (11.8)	15,718 (88.2)	*p* ≤ 0.001
USD 48,001–USD73,000	900 (8.4)	11,136 (91.6)	*p* ≤ 0.001
>USD 73,000	1183 (5.1)	23,126 (94.9)	*p* ≤ 0.001
Prenatal Health Insurance ^†^			
Private	2734 (6.9)	40,919 (93.1)	*p* ≤ 0.001
Medicaid	7435 (19)	33,880 (81)	*p* ≤ 0.001
None	228 (9.2)	1920 (90.8)	*p* ≤ 0.001
Pregnancy Intention			
Intended	3798 (9.6)	37,252 (90.4)	*p* ≤ 0.001
Untended	4951 (13)	34,235 (87)	*p* ≤ 0.001
Unsure	2755 (18.3)	12,829 (81.7)	*p* ≤ 0.001
Parity			
None	3315 (9.3)	33,502 (90.7)	*p* ≤ 0.001
One	3584 (12.5)	26,772 (87.5)	*p* ≤ 0.001
Two or more	4605 (16.1)	24,042 (83.9)	*p* ≤ 0.001

Note: ^†^ Different *n* values for income (*n* = 82,206) and insurance status (*n* = 87,116); ^‡^ denotes significance at *p* ≤ 0.001 across rows.

**Table 4 healthcare-11-01691-t004:** Interaction using logistic regression (*n* = 75,419).

	Adjusted Odds Ratio (95% Confidence Interval)				
	Any Stress (Model 1)	Partner (Model 2)	Trauma (Model 3)	Financial (Model 4)	Emotional (Model 5)
Stressor OR (CI)	1.07 (0.97–1.17)	1.05 (0.96–1.15)	1.07 (0.96–1.18)	1.13 * (1.03–1.24)	0.98 (0.90–1.07)
Race/Ethnicity					
Black	0.70 ^ (0.61–0.81)	0.71 ^ (0.61–0.82)	0.71 ^ (0.61–0.82)	0.71 ^ (0.62–0.82)	0.70 ^ (0.61–0.81)
Hispanic	2.81 ^ (2.32–3.41)	2.82 ^ (2.33–3.41)	2.82 ^ (2.33–3.42)	2.84 ^ (2.34–3.44)	2.81 ^ (2.32–3.41)
American Indian /Alaskan Native	1.43 (0.78–2.64)	1.43 (0.78–2.65)	1.44 (0.78–2.65)	1.44 (0.78–2.66)	1.43 (0.78–2.64)
Asian	1.74 ^ (1.30–2.32)	1.74 ^ (1.30–2.32)	1.74 ^ (1.30–2.32)	1.74 ^ (1.30–2.33)	1.74 ^ (1.30–2.32)
Other	3.36 * (1.50–7.5)	3.36 * (1.50–7.5)	3.36 * (1.51–7.52)	3.38 * (1.51–7.56)	3.36 * (1.50–7.5)
Stressor and Race /Ethnicity Interaction					
Yes Stress and Black	1.27 * (1.07–1.51)	1.32 ^ (1.14–1.54)	1.34 ^ (1.16–1.55)	1.26 * (1.08–1.47)	1.38 ^ (1.18–1.6)
Yes Stress and White	N/A	1.05 (0.95–1.16)	1.05 (0.95–1.16)	1.01 (0.91–1.11)	1.08 (0.98–1.2)
Yes Stress and Hispanic	0.92 (0.72–1.17)	0.96 (0.77–1.21)	0.97 (0.78–1.22)	0.92 (0.73–1.16)	0.99 (0.79–1.25)
Yes Stress and American Indian/Alaskan Native	1.50 (0.75–3.01)	1.56 (0.78–3.13)	1.56 (0.78–3.12)	1.49 (0.75–2.99)	1.62 (0.81–3.25)
Yes Stress and Asian	0.85 (0.55–1.31)	0.88 (0.58–1.35)	0.90 (0.59–1.37)	0.85 (0.56–1.3)	0.91 (0.6–1.4)
Yes Stress and Other	0.50 (0.19–1.3)	0.53 (0.20–1.37)	0.52 (0.20–1.36)	0.50 (0.19–1.29)	0.54 (0.21–1.39)

Note: * Significance at *p* ≤ 0.01; ^ Significance at *p* ≤ 0.001; All models are adjusted for all covariates (maternal age, education, marital status, race/ethnicity, household income, prenatal health insurance, parity, and pregnancy intention).

## Data Availability

Restrictions apply to the availability of these data. Data was obtained from the Centers for Disease Control and Prevention (CDC) Pregnancy Risk Assessment Monitoring System and are available by application at [38] with the permission of federal and state grantees.

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
