# Peer review of "Investigating Maternal Stress, Depression, and Breastfeeding: A Pregnancy Risk Assessment Monitoring System (2016–2019) Analysis"

_healthcare, 2023, doi:10.3390/healthcare11121691_

Round 1

Reviewer 1 Report

The introduction was well written and a joy to read.

lines 54-55: Does the act of seeking preventive healthcare increase or decrease?

I suggest citing the latest version of DSM (DSM-5- TR (Line 58).

Over 70,000 cases were deleted due to missing data from stressful variables and another 38,159 cases were deleted due answering "do not known/not sure", etc. for all of the variables included in the study. This should be addressed in more depth in the discussion. It is possible that those who declined to respond to stressful variables or the other variables may be different than those who responded. 

The states that were represented in the study seemed a good mix of the states. States from different parts of the country are represented. 

Lines 131 - 134: The authors say 'such questions are the following:" Were there more questions? If the respondents answered 'always' or often for one question, but "never", "rarely", or "sometimes", were they coded as "no" or "yes"?

Line 141: How was significance determined for the factors? What analysis was used? What level of significance was used? Was the family error rate controlled?

Lines 145 - 151: Many analyses were performed on the same data. Was the family error rate controlled to reduce the risk of committing a Type 1 error?

Line 154: The authors indicate that bivariate associations are shown in Table 1. Associations infer that coefficients were calculated, but the table contains frequencies and percentages. 

Line 161: Were the percentage of education levels significantly different?

Lines 161 - 165:  The proportion of respondents in the levels of the demographic variables are not proportional. Therefore, respondents in the groups with a greater sample size are more likely to have a greater percentage who report stressors.  

Table1: The percentage symbol is provided only in the first row. However, providing the percentage symbols do help with the interpretation of the table. 

Table 1, Maternal Education, > high school diploma, first and second rows, second column: The percentages (9) and (26) are the only ones without a decimal place. In the first row, the percentage is 9.7% instead of 9%. 

Tables 1-3: I wonder if the percentages are adversely affected by the way they are calculated. The calculations may be making the groups with more respondents look like they have a greater percentage who responded to the item. For example, take Table 1, <High school diploma and Partner Stress (10.2%). Instead of using all respondents who answered the Partner Stress item, should the percentage be based on the number of respondents in the <high school diploma (# of yes to partner stress / # of respondents who answered to partner stress).

Lines 190 - 193: How do these percentages compare to national percentages? Is this sample similar to national levels? If not, I suggest discussing in the discussion. Also, if the data does not reflect the make up of the population, there may be a sampling bias and it should be corrected. 

Line 195: The percentage for the Hispanic groups seems to be missing. 

Pages 10 - 13: The results are reported in both the text and the tables. Normally, the results are reported in one, not both. The results in the text makes reading the sections challenging. The exception is Section 3.6, in which the highlights of the results are discussed in a manner that is easy to read. It seems that a different author wrote this section. Consider making all of the sections like this one. 

Table 6: The Non-Hispanic White group is the reference group. I assume this group was assigned as the reference group because it was the largest one. However, I wonder if this sends the message that the Non-Hispanic White is the standard to compare everything to in the United States. In other words, is this considered the "normal" group? I am not sure how to fix this possible problem, but you may want to address it in the data analysis section.

Line 342: Should it be Table 8 instead of Table 7?

Lines 344 - 346: It seems that the values are different than the ones in the table. For example, the partner-related result in the text is reported as any stressor in the table. 

Line 354: Black should be capitalized.

Line 356: White should be capitalized.

Table 8: Is No Stressor supposed to be in Table 8?

In the first two paragraphs of the discussion, there seems to be discussion about results from other studies than this study. I suggest expanding your discussion about your results in these paragraphs. 

Line 447: Instead of  "on the server" consider changing it to "in the dataset".

Lines 429 - 431: Are there interventions that would be effective? Consider citing some or some methods providers can use to avoid traumatizing Black women.

Author Response

Reviewer 1’s Point-by-Point Response

  • The introduction was well written and a joy to read.
  • lines 54-55: Does the act of seeking preventive healthcare increase or decrease?
    • Added clarifying word to make this distinction
  • I suggest citing the latest version of DSM (DSM-5- TR (Line 58).
    • Changed citation to the latest version of DSM (DSM-5- TR).
  • Over 70,000 cases were deleted due to missing data from stressful variables and another 38,159 cases were deleted due answering "do not known/not sure", etc. for all of the variables included in the study. This should be addressed in more depth in the discussion. It is possible that those who declined to respond to stressful variables or the other variables may be different than those who responded.
    • I will be sure to add a limitation on this in the discussion, but I corrected this section.
  • The states that were represented in the study seemed a good mix of the states. States from different parts of the country are represented.
    • Thank you.
  • Lines 131 - 134: The authors say 'such questions are the following:" Were there more questions? If the respondents answered 'always' or often for one question, but "never", "rarely", or "sometimes", were they coded as "no" or "yes"?
    • They were coded as yes or no—I fixed the wording on this for clarity.
  • Line 141: How was significance determined for the factors? What analysis was used? What level of significance was used? Was the family error rate controlled?
    • Through chi-square tests—fixed in manuscript.
  • Lines 145 - 151: Many analyses were performed on the same data. Was the family error rate controlled to reduce the risk of committing a Type 1 error?
    • In order to combat against this, we changed the significance to 0.01.
  • Line 154: The authors indicate that bivariate associations are shown in Table 1. Associations infer that coefficients were calculated, but the table contains frequencies and percentages.
    • Corrected to change the language on this.
  • Line 161: Were the percentage of education levels significantly different?
    • Yes—all
  • Lines 161 - 165: The proportion of respondents in the levels of the demographic variables are not proportional. Therefore, respondents in the groups with a greater sample size are more likely to have a greater percentage who report stressors.
  • This was re-done.
  • Table1: The percentage symbol is provided only in the first row. However, providing the percentage symbols do help with the interpretation of the table.
    •  
  • Table 1, Maternal Education, > high school diploma, first and second rows, second column: The percentages (9) and (26) are the only ones without a decimal place. In the first row, the percentage is 9.7% instead of 9%.
    •  
  • Tables 1-3: I wonder if the percentages are adversely affected by the way they are calculated. The calculations may be making the groups with more respondents look like they have a greater percentage who responded to the item. For example, take Table 1, <High school diploma and Partner Stress (10.2%). Instead of using all respondents who answered the Partner Stress item, should the percentage be based on the number of respondents in the <high school diploma (# of yes to partner stress / # of respondents who answered to partner stress).
  • Lines 190 - 193: How do these percentages compare to national percentages? Is this sample similar to national levels? If not, I suggest discussing in the discussion. Also, if the data does not reflect the make up of the population, there may be a sampling bias and it should be corrected.
    • Our percentages are very similar to national breastfeeding average estimates.
  • Line 195: The percentage for the Hispanic groups seems to be missing.
    •  
  • Pages 10 - 13: The results are reported in both the text and the tables. Normally, the results are reported in one, not both. The results in the text makes reading the sections challenging. The exception is Section 3.6, in which the highlights of the results are discussed in a manner that is easy to read. It seems that a different author wrote this section. Consider making all of the sections like this one.
    • This was corrected and cut down significantly.
  • Table 6: The Non-Hispanic White group is the reference group. I assume this group was assigned as the reference group because it was the largest one. However, I wonder if this sends the message that the Non-Hispanic White is the standard to compare everything to in the United States. In other words, is this considered the "normal" group? I am not sure how to fix this possible problem, but you may want to address it in the data analysis section.
    • This group is generally what we saw being compared to in the literature we reviewed. Because the US has comparable Black-White disparities, especially in the healthcare industry, we decided to use non-Hispanic White race/ethnicity as the reference.
  • Line 342: Should it be Table 8 instead of Table 7?
    • Corrected
  • Lines 344 - 346: It seems that the values are different than the ones in the table. For example, the partner-related result in the text is reported as any stressor in the table.
    •  
  • Line 354: Black should be capitalized.
    •  
  • Line 356: White should be capitalized.
    •  
  • Table 8: Is No Stressor supposed to be in Table 8?
  • - Corrected
  • In the first two paragraphs of the discussion, there seems to be discussion about results from other studies than this study. I suggest expanding your discussion about your results in these paragraphs.
    •  
  • Line 447: Instead of "on the server" consider changing it to "in the dataset".
    •  
  • Lines 429 - 431: Are there interventions that would be effective? Consider citing some or some methods providers can use to avoid traumatizing Black women.
    • One intervention is early screening.

Reviewer 2 Report

This manuscript addresses an important relationship, between maternal stress and breastfeeding and uses a large nationally representative data set to investigate this relationship. 

It is unclear how the main outcome of breast-feeding (based on two questions) is defined. Is it any positive answer to either question? Unfortunately, this measure of breast-feeding is very minimal since it could include a single attempt to breast feed. If the "current" breast-feeding question was positive, that would be more useful if the time in the postpartum period was known. Without this information, the main outcome doesn't have much information about breast-feeding.

I was very concerned about the handling of missing data, since the original sample was over 202,000 and the final sample was 94,018 - in other words, more than half of the sample was excluded for missing data and no attempt was made to understand the mechanism of missingness. Listwise deletion assumes the data are missing completely at random which seems very unlikely. 

There weren't enough details in the methods to understand how variables were created. For instance, for postpartum depression, how many questions were involved and how were the responses combined - was it an average? Similarly, how were multiple stressors in each category combined, how were covariates coded. It is not possible to evaluate proper methodology without understanding these items.

The interpretation of Table 1 is incorrect. For instance, you state that women between the ages of 25 and 34 years old indicated higher rates of experiencing any of the 13 stressors - however, the rate of 58.7% is actually lower than the representation of women of that age in the study (60.7%). The table was very difficult to interpret since the headings were not reproduced on the multiple pages so it was hard to make sense of what each column represented. 

In Table 2, you also need to give the percentages of women in each category (age group, education group), so that it is possible to see which categories were overrepresented in the depression results. Again, the interpretation should be based on whether mothers were overrepresented in each category - for instance, in the sample,21% of mothers were less than 24 years old but 30% of those with prenatal depression were <=24 years old so that category is overrepresented in prenatal depression. 

I don't think it makes sense to report the unadjusted ORs throughout the paper. If you have covariates that are affecting the results, then what is interesting are the adjust ORs and reporting both takes up a lot of space without adding useful information. 

Table 8, I can't make sense of the column which has nothing for the first 5 rows and  "Yes" in the rest of the rows. Can you clarify what that means. 

"untended" should be "unintended"

Author Response

Reviewer 2’s Point-by-Point Response

This manuscript addresses an important relationship, between maternal stress and breastfeeding and uses a large nationally representative data set to investigate this relationship.

It is unclear how the main outcome of breast-feeding (based on two questions) is defined. Is it any positive answer to either question? Unfortunately, this measure of breast-feeding is very minimal since it could include a single attempt to breast feed. If the "current" breast-feeding question was positive, that would be more useful if the time in the postpartum period was known. Without this information, the main outcome doesn't have much information about breast-feeding.

  • These PRAMS questions were used by previous breastfeeding studies, but we added more details for this to clarify the coding.

I was very concerned about the handling of missing data, since the original sample was over 202,000 and the final sample was 94,018 - in other words, more than half of the sample was excluded for missing data and no attempt was made to understand the mechanism of missingness. Listwise deletion assumes the data are missing completely at random which seems very unlikely.

  • Corrected—To address the issue with missing variables for insurance and income we performed sensitivity analyses to compare the estimates of our outcome variables when we imputed the missing observations. We used the mlogit function to impute insurance cat with the following predictors. We will used the ologit function to impute income cat with the following predictors. Then we used the imputed variables in our regression models as predictors. The findings are shown in table appendices. We found that there was no significant change in the aOR estimates of the association between your predictor variables and outcome variable (breastfeeding).

There weren't enough details in the methods to understand how variables were created. For instance, for postpartum depression, how many questions were involved and how were the responses combined - was it an average? Similarly, how were multiple stressors in each category combined, how were covariates coded. It is not possible to evaluate proper methodology without understanding these items.

  • We used low cutoff p-value at 0.01 (Bonferroni correction).

The interpretation of Table 1 is incorrect. For instance, you state that women between the ages of 25 and 34 years old indicated higher rates of experiencing any of the 13 stressors - however, the rate of 58.7% is actually lower than the representation of women of that age in the study (60.7%). The table was very difficult to interpret since the headings were not reproduced on the multiple pages so it was hard to make sense of what each column represented.

            Corrected.

In Table 2, you also need to give the percentages of women in each category (age group, education group), so that it is possible to see which categories were overrepresented in the depression results. Again, the interpretation should be based on whether mothers were overrepresented in each category - for instance, in the sample,21% of mothers were less than 24 years old but 30% of those with prenatal depression were <=24 years old so that category is overrepresented in prenatal depression.

  •  

I don't think it makes sense to report the unadjusted ORs throughout the paper. If you have covariates that are affecting the results, then what is interesting are the adjust ORs and reporting both takes up a lot of space without adding useful information.

  • Will not include unadjusted ratios and corrected.

Table 8, I can't make sense of the column which has nothing for the first 5 rows and  "Yes" in the rest of the rows. Can you clarify what that means.

  • Table 8 is being re-done. Corrected.

Round 2

Reviewer 1 Report

The results section is much easier to read and is more meaningful. Emphasizing the important results allows the reader to compare and contrast the findings among the different sections. The tables also are easier to read. The reader can easily read the results and see how they apply to the information in the discussion. 

Please consider addressing the lost respondents in the limitation section. Please see Line 112. It is reported that 30% were lost because data was not available from Question 19. This differs from the income and insurance that was missing (see Lines 426-427). 

Lines 149 and 150: ..prenatal depression and postpartum depression.

Lines 165 and 166: If the respondents answered “yes” to one question and “no” to the other, was it considered “no” for postpartum depression?

Lines 224-225: I am not sure what is being said in the sentence. Is there a word missing?

Consider making Figures 1 and 2 the same size.

Author Response

The results section is much easier to read and is more meaningful. Emphasizing the important results allows the reader to compare and contrast the findings among the different sections. The tables also are easier to read. The reader can easily read the results and see how they apply to the information in the discussion. 

Please consider addressing the lost respondents in the limitation section. Please see Line 112. It is reported that 30% were lost because data was not available from Question 19. This differs from the income and insurance that was missing (see Lines 426-427). 

- added sentence in limitations section: sites that did not include our exposure or outcome questions from the PRAMS questionnaire were excluded

Lines 149 and 150: ..prenatal depression and postpartum depression.

- not sure what to fix here, but re-wrote the section. 

Lines 165 and 166: If the respondents answered “yes” to one question and “no” to the other, was it considered “no” for postpartum depression?

- Wrote to be marked as yes for postpartum depression, both questions needed to have been marked yes. 

Lines 224-225: I am not sure what is being said in the sentence. Is there a word missing?

- Fixed. 

Consider making Figures 1 and 2 the same size.

- Fixed. 

Reviewer 2 Report

Here are my comments on the revision:

Some changes are highlighted and some are not – this is confusing. 

Point 1 -Did not clarify breastfeeding questions. “yes” both and “no” to both are covered – what about one “yes” and one “no”? 

Missing data – not sure how mlogit function does imputation – no discussion of how imputation was done in manuscript of appendix. I cannot understand how you did the imputation and therefore cannot report on its accuracy.  

Coding of variables  - not clarified 

Misinterpretation of table 1 – not fixed 

Table 2 – not addressed 

Table 8 (now table 4) I still cannot make sense of this table, nor do I understand how the interaction was modelled.

no

Author Response

Some changes are highlighted and some are not – this is confusing. 

Point 1 -Did not clarify breastfeeding questions. “yes” both and “no” to both are covered – what about one “yes” and one “no”? 

- Participants who answered yes to only one of two questions were coded as breastfeeding as well. 

Missing data – not sure how mlogit function does imputation – no discussion of how imputation was done in manuscript of appendix. I cannot understand how you did the imputation and therefore cannot report on its accuracy.  

- We used the ologit function to impute income missingness and the mlogit function to impute insurance missingness with the raw site and covariate variables as predictors. We used the imputed variables in our regression models as predictors, and the findings are shown in appendix. 

Coding of variables  - not clarified 

- I believe the coding of variables is very well detailed in the methods section. 

Misinterpretation of table 1 – not fixed -- Based on previous papers, this interpretation is of the descriptive frequencies, and we conducted Chi-square tests to indicate significant differences across the column tests. Authors are not quite understanding how the interpretation is incorrect. 

Table 2 – not addressed -- rewrote this section, and the variable depression is not one variable, the questions asking about preconception, prenatal, and postpartum depression were converted into variables so each one is composed of the N #, but the table only shows the yes %. 

- Table 8 (now table 4) I still cannot make sense of this table, nor do I understand how the interaction was modelled.

  • used the following code for the model: foreach var of varlist strs pstrs tstrs fstrs estrs{
    svy, subpop(if subpop_var==1): logistic brstfeed i.`var' ib2.racethn i.strs#i.racethn i.agecat i.educat i.married i.income i.inscat i.intent i.parity  
    }
  • The interaction ran analyses of the stress x breastfeeding, race/ethnicity x breastfeeding, and then ran stress x race/ethnicity with breastfeeding. There are five models which represents each logistic regression interaction performed and their results.